# Species Diversity and Community Composition of Macroinvertebrates in Headwater Streams of Two Subtropical Neighboring Lowland Basins

**Lu Wang** [1], **Xiaochen Lv** [1], **Jiaxu Li** [1], **Lin Tan** [1], **Eric Zeus Rizo** [2] and **Bo-Ping Han** [1,*]

[1] Department of Ecology and Institute of Hydrobiology, Jinan University, Guangzhou 510632, China; 15590689772@163.com (L.W.); 17853729054@163.com (X.L.); brendaljx@126.com (J.L.); tanlinwangyi@163.com (L.T.)

[2] Division of Biological Sciences, College of Arts and Sciences, University of the Philippines-Visayas, Miagao, Iloilo 065023, Philippines; ecrizo@up.edu.ph

[*] Correspondence: tbphan@jnu.edu.cn or tbphan@126.com

**Abstract:** Determining the patterns of diversity and community composition in headwater streams is fundamental to river biodiversity conservation. Environmental selection has been assumed to be a major force driving temperate macroinvertebrate assembly. To test this assumption in the subtropics, we conducted identical surveys for headwater streams in two neighboring basins, which are located on two sides of a north–south mountain in southern China. We sampled macroinvertebrates and measured habitat and environmental variables in headwater streams of the two basins. The two groups of headwater streams share a species pool and have similar taxa, taxa richness, and functional composition. Beta diversity accounted for a high proportion of the within-basin diversity, and was mainly attributed to the replacement differences. Highly similarity between the two basins' species composition showed similar environmental selection in structuring macroinvertebrate communities at the regional scale. Redundancy analysis showed that basin identity is the key factor explaining the variation of communities. Environmental selection is an important factor in structuring macroinvertebrate communities within individual basins. Annual precipitation differs significantly on the two sides of the mountain shaded by the East Asia monsoon, resulting in distinctive substrate compositions in the two basins. Our study provides empirical support for the roles of environmental selection in shaping subtropical macroinvertebrate communities.

**Keywords:** macroinvertebrate metacommunity; beta diversity; environmental selection; spatial process; RDA

## 1. Introduction

Community composition and the underlying mechanisms of species diversity patterns are fundamental topics in community ecology [1–3]. A metacommunity refers to a set of local communities in a given region that are potentially connected by species dispersal [4–6]. The theoretical framework for metacommunities provides a way to evaluate how local (e.g., environmental heterogeneity) and regional processes (e.g., dispersal limitation) mediate the assembly processes of natural communities, and how spatial patterns govern beta diversity between local communities [6]. Beta diversity is influenced by multiple ecological processes, and can be decomposed into two components (e.g., species replacement or turnover, richness differences or nestedness) [7–9]. The relative contributions of these components to beta diversity may be related to the partitioning of environmental resources among species [10,11]. As beta diversity links local and regional diversity, knowing the mechanisms underlying its patterns helps us identify processes driving the patterns of species diversity at multiple spatial scales [1,12,13].

River ecosystems drain the landscape through hierarchical series of fluvial channels and are rather dynamic with spatial heterogeneity. Such ecosystems all begin with small headwater streams, which are not only highly heterogeneous but also isolated from each other [14,15]. Headwater streams offer an amenable model system for studying community variation in a metacommunity context. Such streams are characterized by high richness of macroinvertebrates [16,17], which play an important ecological role in the material cycling and energy transfer of river ecosystems [16,18]. Unraveling macroinvertebrate diversity and potential structuring mechanisms at the regional scale are fundamental to stream monitoring, health assessment, and ecological restoration.

Our knowledge of stream macroinvertebrate assemblages is mostly from temperate regions, especially in Europe and America [19–21]. Most stream studies have found that environmental selection and ecological drift prevail over the spatial process in shaping the community composition of macroinvertebrates at the local scale [22,23]. Roles of spatial processes (e.g., dispersal limitation) and environmental selection (e.g., environmental filtering) increase at regional scale [24,25]. In the tropics and subtropics, environmental conditions (e.g., higher temperature, constant climatic history, more diverse predators, and substantial hydrological fluctuations) support a higher species diversity of macroinvertebrates [26,27]. Ecological drift (i.e., change in species population size due to random births and deaths) may have a greater influence on the diversity and composition of macroinvertebrate communities in such warmer climates [28,29]. Recent studies have shown that environmental selection also plays a key role in the macroinvertebrate assembly of tropical and subtropical streams [30,31]. Thus, there is a need for more relevant research in warmer areas to place both tropical and temperate systems into a global context.

Here, we focused on headwater streams of the Beijiang River and Dongjiang River, two large subtropical and lowland tributaries of the Pearl River, the largest river in southern China. The Pearl River basin is characterized by a monsoonal climate with two hydrologically contrasting seasons, a flooding season from May to September, and a dry season from October to April. As the two basins are in close proximity and cover the same latitudinal zone, they are expected to have similar habitat conditions and share a species pool. Thus, we hypothesize that spatial processes (dispersal limitation) and environmental selection (filtering) are similar between the basins, and environmental selection at the local scale is the major factor shaping macroinvertebrate communities. As a consequence, there will be similar species composition and community structure in headwater streams for the two neighboring basins.

We conducted identical surveys of headwater stream macroinvertebrates in the Beijiang basin and the Dongjiang basin region. The two groups of headwater streams are located on two sides of a north–south mountain. Such locations allow us to easily compare environmental conditions and their roles. We analyzed the taxa composition, local diversity, and beta diversity of macroinvertebrate communities and examined the effects of regional (environmental filters, dispersal limitation, and basin identity) and local ecological processes (environmental selection) on community structure. The study will improve our knowledge and understanding of macroinvertebrate assemblages in headwater streams of tropical and subtropical Asia.

## 2. Materials and Methods

### 2.1. Study Area

The Pearl River is the second-largest river in China, which covers a region of subtropical to tropical monsoon climate straddling the Tropic of Cancer. The Pearl River is composed of three major rivers: Xijiang, Beijiang, and Dongjiang (Figure 1). Xijiang is the largest branch originating in the Yunnan-Guizhou Plateau, flowing from west to east. The Beijiang and Dongjiang rivers are located in the eastern lowland, flowing from north to south and merging with the Xijiang river into the Zhujiang Pearl River Delta. This study was conducted in the headwater streams of the Beijiang Basin and the Dongjiang basin. Together, we investigated seventeen headwater streams in the two basins, including 9 streams

at altitudes of 310–690 m in the Beijiang basin and 8 streams at altitudes of 370–928 m in the Dongjiang basin. This study focused on undisturbed headwater streams and as such the sampled streams were the only ones we could find. The two groups of headwater streams are located on two sides of Mountain Jiulian, which extends northwards from Jiangxi Province to Guangdong Province in southern China. The headwater streams of Dongjinag basin are located on the east side of Mountain Jiulian, which peaks at about 1400 m above sea level, and the headwater streams of the Beijiang basin at the west side. The two groups of headwater streams are about 128 km distant from each other and have quite similar air temperature, soil, and vegetation (subtropical evergreen broad-leaved forests).

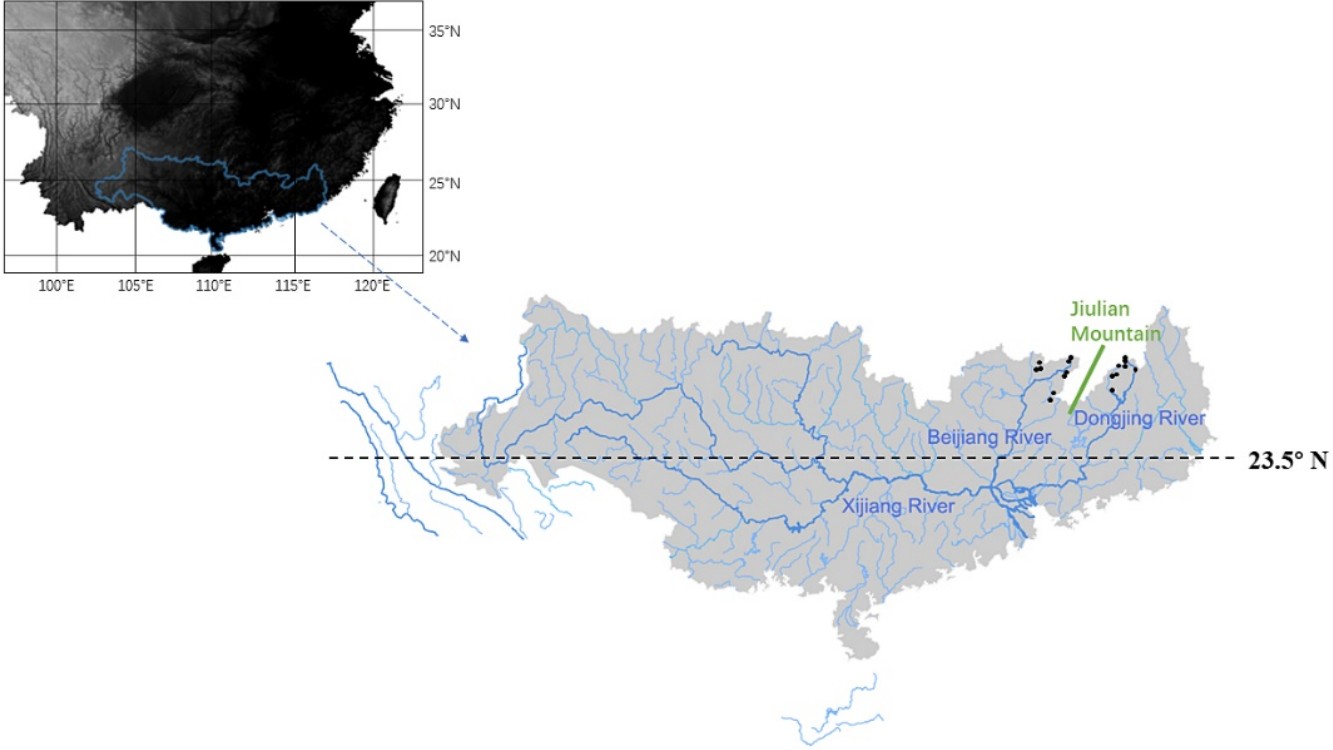

**Figure 1.** The Pearl River Basin and the distribution of 17 sampling sites in the study area. The two groups of headwater streams are located on two sides of Mountain Jiulian, including 9 streams at altitudes of 310–690 m in the Beijiang basin (the west side of Mountain Jiulian) and 8 streams at altitudes of 370–928 m in the Dongjiang basin (the east side of the Mountain Jiulian).

*2.2. Sampling and Identification of Macroinvertebrates*

Macroinvertebrates were collected in winter, from 16 January to 27 January 2021, when the hydrological disturbance was seasonally low. At each sampling site, macroinvertebrates were obtained from five quantitative replicates using a Surber sampler (30 × 30 cm, 500 μm mesh). The five microhabitats (three riffles and two pools) were sampled to cover the most representative microhabitats along a 100 m reach of each site. All the organisms were hand-separated from detritus and stored in 75% ethanol. Macroinvertebrates were identified to the family level, excluding Chironomidae. As Chironomidae contains an extraordinarily high number of species, it was identified to the level of sub-family. Identification and determination of taxa abundance were performed under a stereoscopic microscope according to the available literature [32–34]. They were then grouped into functional feeding groups, based on morphological and behavioral adaptations to acquire their food resources [35,36]. These functional groups were gather-collectors, filter-collectors, scrapers, shredders, and predators.

#### 2.3. Measurements of Environmental Variables

When macroinvertebrates were sampled, habitat and environmental variables were measured at each site in the field. Water samples were also brought back to the laboratory for variables that cannot be measured in situ. Reach width (Wid) was measured as the average of three equidistant transects (both sides and middle of the 100 m sampling reach) by a laser rangefinder. The water velocity (Vel) was measured at four to eight points along three cross-sections by a portable velocity analyzer. The water depth (Dep) was measured as the average of three evenly spaced points along transects using a graduated stick. The water temperature (Temp), pH, conductivity (Cond), dissolved oxygen (DO), mmHg, and ORP were measured using a portable water quality analyzer (YSI). Water samples were collected and maintained at 4 °C in a cryogenic box and transported to the laboratory for analysis within 48 h. Chemical variables included total nitrogen (TN), nitrate ($NO_3$), nitrite ($NO_2$), ammonium ($NH_4$), total phosphorus (TP), phosphate ($PO_4$), silicon dioxide ($SiO_2$), and chlorophyll (Chl) were measured according to standard methods [37].

The substrates of each sampling site were described by visually estimating the percentages of boulders, cobble, pebble, gravel, sand, and silt following the established protocol by Cummins [38]. Cover% (proportion of riparian cover) and detritus% (leaf litter cover) were estimated by visual inspection.

Elevation and bioclimatic variables (AP: Annual Precipitation and AMT: Annual Mean Temperature) were extracted from the WorldClim database [39].

#### 2.4. Data Analyses

To reveal the potential regional environmental difference, each environmental variable between the Beijiang streams and Dongjiang streams was compared using the Mann–Whitney U test in SPSS 21.0 software, because the results of the normality test showed that the environmental variables do not normally distributed. Principal components analysis (PCA) was performed to describe chemical, physical, and climatic variables of streams by synthesizing multivariable information into two dimensions using the prcomp function in R v4.0.3. To avoid co-linearity and overfitting the data, we further reduced the number of chemical and physical variables as a parsimonious combination of environmental variables. By performing principal components analysis (PCA), the chemical variables were reduced to the first two principal components (PC1 and PC2) as explanatory variables in redundancy analysis (RDA). The correlations between the first two principal components and chemical factors are shown in Table S1. The substrate composition was described by calculating the average substrate score (MSUBSTD) [40] as follows:

$$\text{MSUBSTD} = \frac{-7.75 \times \text{BOLDCOBB} - 3.25 \times \text{PEBBGRAV} + 2 \times \text{SAND} + 8 \times \text{SILTCLAY}}{\text{TOTSUB}}$$

$$\text{TOTSUB} = \text{BOLDCOBB} + \text{PEBBGRAV} + \text{SAND} + \text{SILTCLAY}$$

where BOLDCOBB, PEBBGRAV, SAND, and SILTCLAY indicate the percentage cover of bolder/cobble, pebble/gravel, sand, and silt/clay, respectively. A higher score indicates higher proportions of sand and silt, whereas a lower score indicates a higher proportion of large rocks and cobble.

We calculated taxa richness and total density for all macroinvertebrates and each functional feeding group at each location. T-test was used to detect the differences in local diversity between the two basins in SPSS v21.0 software, because the data conform to normality and homogeneity of variance. Principal coordinate analysis (PCoA) was performed to identify the differences in taxonomic composition between the two basins, based on Bray–Curtis dissimilarity. A nonparametric permutational multivariate analysis of variance (PERMANOVA) with 999 permutations was employed to examine whether the taxa composition of macroinvertebrates significantly differed between the two basins. PCoA and PERMANOVA analyses were conducted with the vegan package in R v4.0.3.

We used the additive partitioning approach ($\gamma = \alpha + \beta$) [41,42] to estimate beta diversity in each basin. Here, $\alpha$ is the average taxa richness in local communities, while $\gamma$ refers to the total taxa richness observed in the basin. $\beta$ means the variation between multiple streams within a basin. The abundance data was used for the hierarchical analysis of diversity partitioning. Then, we used the Jaccard dissimilarity coefficient based on presence/absence data to decompose beta diversity into replacement ($\beta repl_J$) and richness difference ($\beta rich_J$) [8,43]. We also used the Ružicka dissimilarity coefficient based on abundance data to decompose the total beta diversity into replacement ($\beta repl_R$) and abundance difference ($AbDiff_R$) components [44]. These analyses were conducted with the function beta.div.comp in the adespatial package in R v4.0.3 [9]. T-test was also used to detect the difference in beta diversity between the two basins in SPSS v21.0 software, because the data conform to normality and homogeneity of variance.

We performed an RDA to determine the factors that influence taxa composition and community variation of the macroinvertebrate communities. We constructed three explanatory models: an environmental model, based on the parsimonious combination of measured environmental variables; a basin model, modeled by a dummy variable "basin identity" to represent effects between basins; a spatial model, which described spatial processes within the basins. We used the function create.MEM.model provided by Declerck et al. [23] via the PCNM package in R v4.0.3 to construct the spatial model, which is suitable for nested sampling designs. This function produces a set of orthogonal spatial variables in a staggered matrix divided by groups based on the geographical coordinates, number of groups of sites, and sampling sites in each group. Each group represents the hierarchical spatial distribution of the sampling points and different groups receive a value of zero (0) for each spatial variable created. We obtained a total of 9 spatial vectors (Moran's eigenvector maps, MEMs) in our study. Species abundance was Hellinger transformed prior to RDA. We tested the significance of the full model of RDA (i.e., full variables) using the ANOVA function in the vegan package. Only if the full model was significant, a forward selection procedure was conducted with the ordiR2step function in the vegan package to select the key factors that significantly influence the macroinvertebrate components. The hierarchical partitioning method was used to distinguish a single variable's contribution via the rdacca.hp package in R v4.0.3 [45].

## 3. Results

### 3.1. Environment Factors

Chemical variables, physical variables, and climatic variables all showed some differences between the Beijiang basin and the Dongjiang basin (Figures 2 and S1). Chemical variables such as DO and pH were significantly higher in the Dongjiang basin. MSUBSTD (the substrate composition) was significantly higher in the Beijiang basin. AP (Annual precipitation) was higher in the Dongjiang basin.

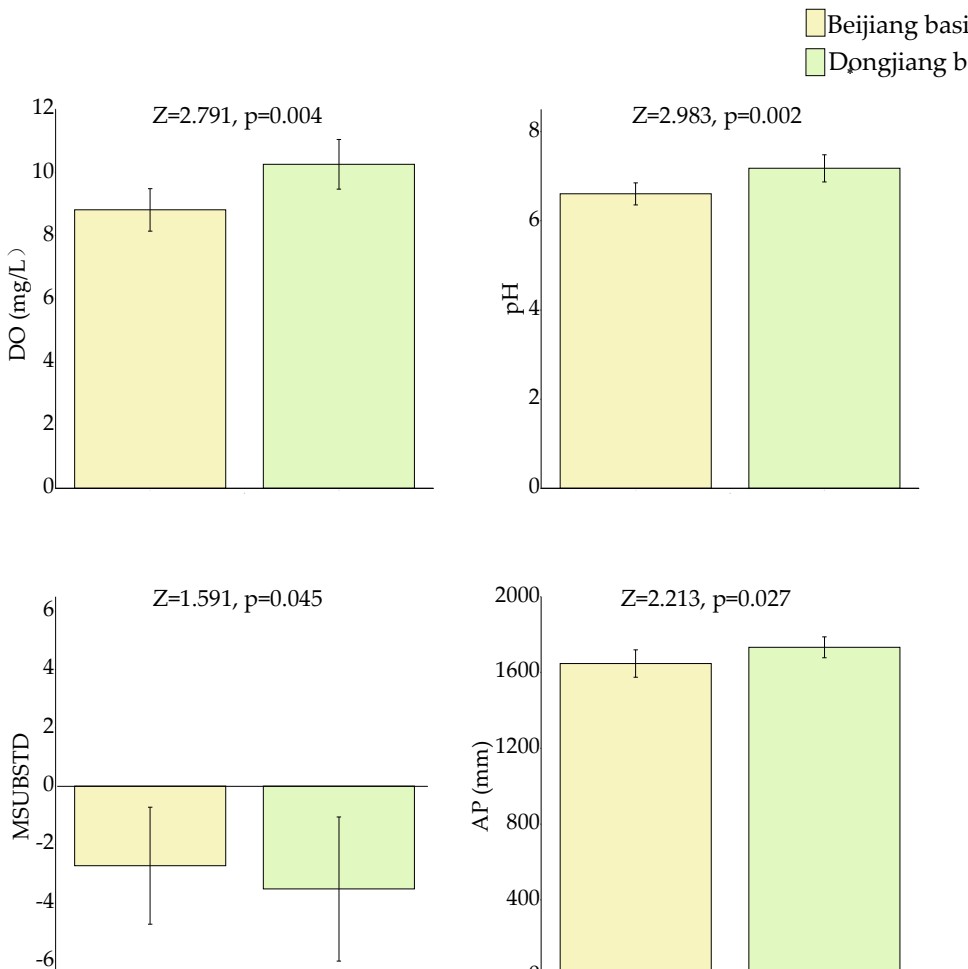

**Figure 2.** Environmental variables (DO, pH, MSUBSTD, and AP) (Means ± SD) with significant differences between two groups of headwater streams in the Beijiang basin and the Dongjiang basin.

### 3.2. Taxa Composition, Alpha and Gamma Diversity

A total of 60 families were identified from 5 phyla and 14 classes. In total, 51 families were found in the Beijiang basin and 45 families in the Dongjiang basin (Table S2). In terms of functional feeding group composition, gather-collectors were the most dominant based on total abundance, followed by filter-collectors, predators, and shredders, while scrapers accounted for a little part. The mean density of total macroinvertebrates and each functional feeding group were not significantly different between the two basins (Table 1).

**Table 1.** The *p* values for *t*-test results of density between the two basins. Mean values ± SD of the density of total macroinvertebrates and each functional feeding group in the Beijiang basin and Dongjiang basin are also shown.

|  | Beijiang Basin | Dongjiang Basin | *p* Value |
| --- | --- | --- | --- |
| Total macroinvertebrate | 823.951 ± 305.155 | 785.556 ± 283.806 | 0.793 |
| Gather-collector | 428.889 ± 191.804 | 397.222 ± 140.700 | 0.707 |
| Filter-collector | 149.63 ± 129.438 | 113.056 ± 92.950 | 0.519 |
| Scraper | 53.086 ± 31.140 | 83.889 ± 67.086 | 0.263 |
| Predator | 100.247 ± 59.695 | 83.333 ± 57.814 | 0.563 |
| Shredder | 92.099 ± 76.369 | 108.056 ± 69.241 | 0.660 |

Baetidae, Ephemeridae, Chironomidae, Hydropsychidae, and Simuliidae were the most common and abundant families in the two basins. PERMANOVA analysis (F = 1.562, $p$ = 0.079) and PCoA (Figure 3) indicated that the community composition was different between the two basins. SIMPER analysis indicated that the difference was mainly (cumulative contribution > 80%) caused by the higher abundance of Baetidae, Ephemeridae, Chironominae, Heptageniidae, Tipulidae, Nemouridae, and Ephemerellidae in the Dongjiang basin, while there was a higher abundance of Orthocladiinae, Simuliidae, Hydropsychidae, Tanypodinae, Siphlonuridae and Leptophlebiidae in the Beijiang basin.

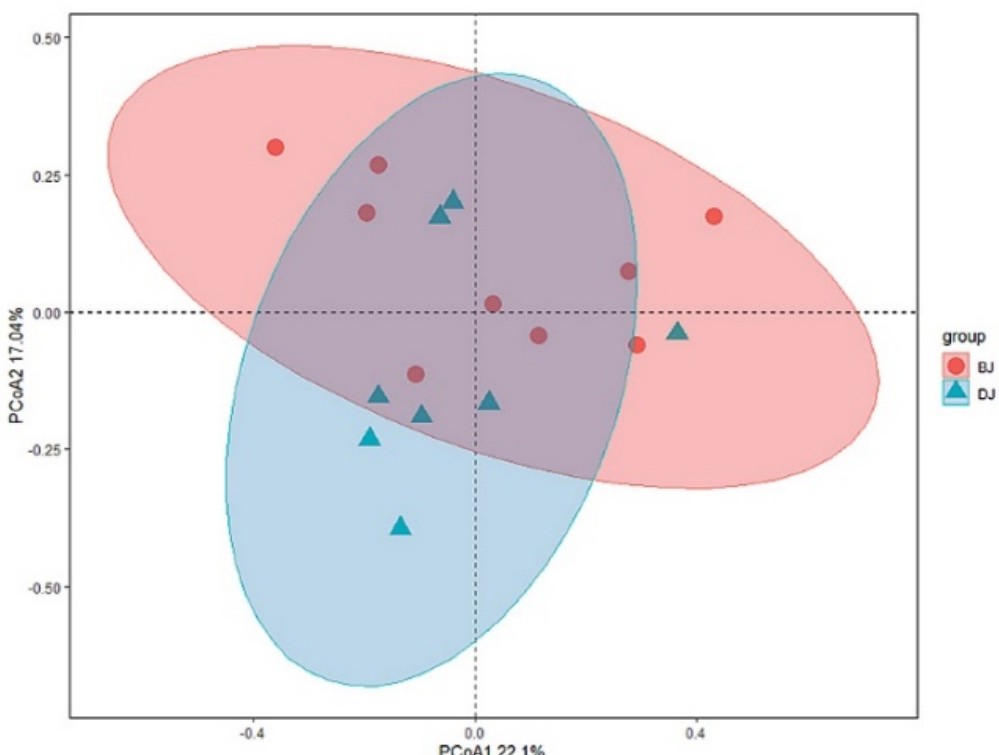

**Figure 3.** Principal coordinate analysis (PCoA) of Beijiang (red) and Dongjiang (blue) communities, in which the community distance was measured in Bray–Curtis dissimilarity. BJ: Beijiang basin; DJ: Dongjiang basin.

### 3.3. Beta Diversity and Its Components

A full hierarchical diversity partitioning of taxa richness showed a substantial contribution of beta diversity to gamma diversity (Figure 4A), contributing 55.56% and 48.61% in the Beijiang basin and Dongjiang basin, respectively. Beta diversity of Jaccard dissimilarity was higher in the Beijiang basin than in the Dongjiang basin ($p$ = 0.000), but Ružicka dissimilarity was similar between the two basins ($p$ = 0.234). With the Jaccard index or Ružicka indices, the replacement difference component showed higher importance to total beta diversity (Figure 4C,D).

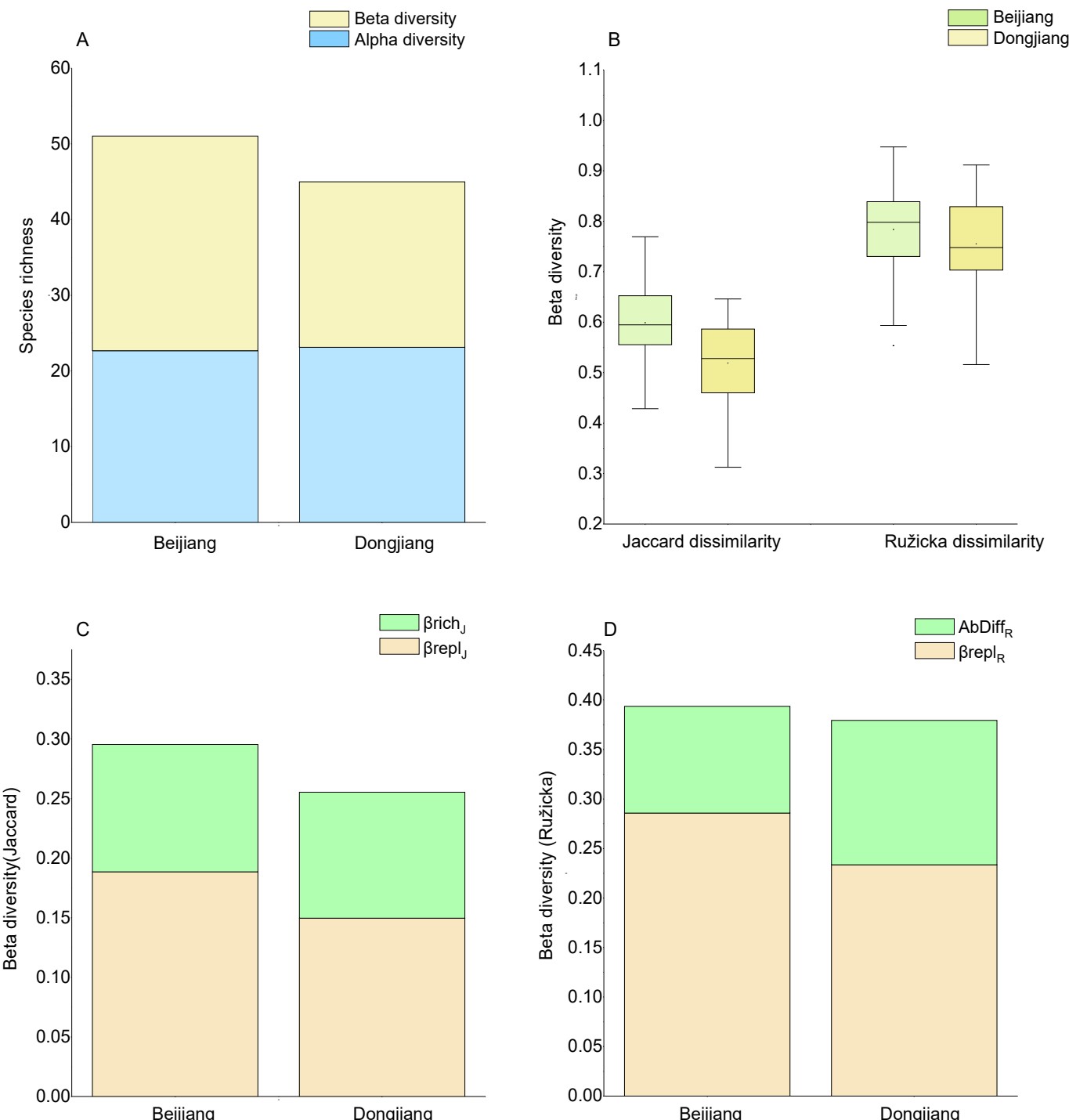

**Figure 4.** Full hierarchical analysis of diversity partitioning for the composition of macroinvertebrates (**A**). Alpha = average stream diversity, beta = diversity among streams. Jaccard index and Ružicka indices in the Beijiang basin and Dongjiang basin (**B**). Mean values of Jaccard dissimilarities (presence and absence data) of macroinvertebrate communities in the Beijiang basin and the Dongjiang basin, decomposed into species replacement ($\beta repl_J$) and richness difference ($\beta rich_J$) component (**C**). Ružicka dissimilarities (abundance data) of macroinvertebrate communities in the Beijiang basin and Dongjiang basin, decomposed into abundance replacement ($\beta repl_R$) and abundance difference (AbDiff$_R$) components (**D**).

### 3.4. Environmental Selection in Structuring Macroinvertebrate Community

RDA showed that the basin identity RDA model (adjR$^2$ = 0.06, $p$ = 0.012) and environmental RDA model (adjR$^2$ = 0.118, $p$ = 0.004) significantly explained the variation in total community structure (Figure 5A), whereas the spatial RDA model was negligible (adjR$^2$ = −0.005, $p$ = 0.506). The forward selection of variables revealed strong evidence supporting the combined effects of chemical, physical, and climate variables in determining the composition of communities (Figure 5B). PC1 (mainly mmHg and Cond, Table S1), MSUBSTD, and AP (Annual Precipitation) were significant drivers of the macroinvertebrate community, explaining 2.39%, 4.63%, and 1.92% of the total variation of the communities, respectively.

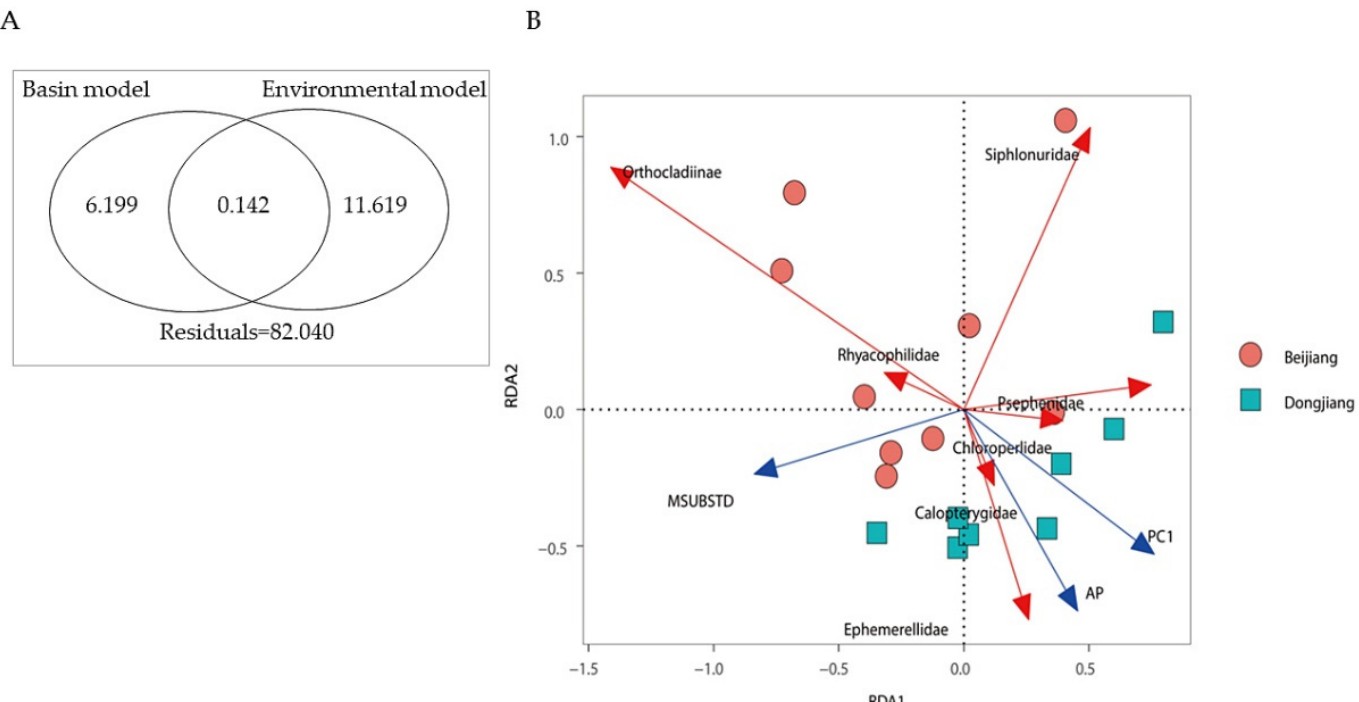

**Figure 5.** Venn diagram illustrating results of variation partitioning for the overall macroinvertebrate metacommunity (**A**) and RDA plot of macroinvertebrates and environmental variables (**B**). Values in the circles indicate the amount of variation in the community composition data explained by the basin model and environmental model. Residuals are shown below the circle. All fractions ($p < 0.05$) are based on adjusted R$^2$ values shown as percentages of the total variation. AP: Annual Precipitation; PC1: First principal components of principal component analysis of chemical factors (see Table S1).

## 4. Discussion

We observed differences in selected environmental variables between the two neighboring basins. These differences are likely influenced by the East Asian monsoons. The nine headwater streams of the Dongjiang basin are located on the east side of the mountain. Compared with those in the Beijiang basin on the west side, these nine streams receive higher precipitation, and they also cover a wider elevation range. The two basins were composed of similar taxa and functional feeding groups, indicating similar influence of environmental filtering on the species pool. Our analysis showed that the environmental selection significantly explained the taxa composition of macroinvertebrate communities in the area.

### 4.1. Local and Regional Diversity of Macroinvertebrates

It is generally assumed that there is a higher species diversity at lower latitudes in many animal taxa. When identification of species is not consistently possible, family richness is used as an approximate surrogate of species richness [46]. Our stream macroin-

vertebrate family richness was generally high (regional diversity: 51 families in the Beijiang basin, 45 families in the Dongjiang basin), which is similar to other reported numbers in low latitude basin (e.g., 44 families in Ecuador lowland basin; 53 families in Brazil basin; 27–50 families in Malaysia basins) [30,46,47]. Not all macroinvertebrate groups have the same latitudinal pattern, but some aquatic insects (mainly Odonata and Coleoptera) usually maintain higher richness in low latitude areas [48–50]. We also recorded higher diversity of Odonata (4 families, 19.61 ind/m$^{-2}$) and Coleoptera (7 families, 115.69 ind/m$^2$) than many reports in boreal streams with almost no Odonata or no Coleoptera [26,51,52].

Stream shredders play a crucial role in the breakdown of allochthonous leaf litter, which is the key process in temperate headwater streams [52]. In contrast, it has been suggested that litter breakdown is driven by microorganisms, and shredders are scarce in low latitude streams [53–55]. Shredders do not adapt to warm climates and do not like tropical leaves as food sources (because of high concentrations of toxic compounds and physical toughness) [53,56,57]. However, we found shredders were not scarce in our streams, with a relative abundance of 11.18% in the Beijiang basin and 13.76% in the Dongjiang basin. Similar observations have been reported in South America, Panama, and Australia [58–60]. The diversity of shredders in tropics and subtropics is worth being further investigated. In addition, it is generally assumed that there are more predators at low latitudes, supported by complex food web needs [61,62]. Indeed, high proportions of predators occurred in our streams (12.16% in the Beijiang basin, 10.61% in the Dongjiang basin), which was higher than many boreal streams (e.g., 6% in Danish streams, 3% in Ecuadorian Páramo streams) [46].

Regional diversity is linked to local diversity by beta diversity [13]. We found high beta diversity both in the Beijiang and the Dongjiang basins, which contributed more than half of the regional diversity (Figure 4A). The beta diversity based on abundance differences and presence/absence data all showed that the replacement component had greater relative importance than the abundance difference (Figure 4C,D). This indicates that a large proportion of taxa in headwater streams is unique to particular streams, i.e., there is habitat heterogeneity [17,63]. The above result also indicates the important role of headwater streams in maintaining the regional diversity at low altitudes.

We found similar dominant taxa in the two basins, and the difference in abundance of dominant taxa is the main reason for the composition differences in macroinvertebrate communities between the two basins. The dominant taxa usually play a key role in structuring communities, influencing the survival and distribution of other species [64]. We also found a little difference in functional feeding group composition between the two basins, indicating similar trophic dynamics. The highly similar functional group composition strongly implies that headwater stream communities of macroinvertebrates are similarly assembled to adapt to similar environmental conditions. Both similar taxa and functional group composition indicate that macroinvertebrate communities found in the two groups of headwater streams share a species pool.

*4.2. Environmental Factors Shaping Macroinvertebrate Communities*

Supported by the RDA analysis, the environmental model significantly explained the variation of the macroinvertebrate communities, but the spatial model was not significant. Environmental variables significantly and independently explained 11.6% of the community variation. This strongly supports that environmental selection is important in shaping macroinvertebrate metacommunity organization at low latitude. Similar observations have also been reported in both tropical and temperate basins [22,23,65], suggesting that similarities in macroinvertebrate assembly prevail. Our results also showed that basin identity was a significant explanatory factor of variation in community structure, which independently explained 6.2% of community variation (Figure 5A). The basin identity is potentially related to basin independency, caused by low hydrological connectivity and isolation by a high mountain ridge [23]. It also indirectly relates to historical effects and climatic forcing on local community structure [23,66,67]. This effect cannot be directly

measured in our present study and needs biological data containing historical information such as genetics and phylogeny.

The environmental model and basin model (basin identity) only slightly shared 0.142% of explained variation (Figure 5A), implying that similar environmental variables between the basins are more important in shaping the community composition at the family level. Like many studies of headwater streams [68,69], substrate composition (MSUBSTD) was the most important environmental factor in our macroinvertebrate community. Substrates with large particle sizes could provide habitats better for clingers (e.g., Chloroperlidae and Psephenidae) and swimmers (e.g., Siphlonuridae), which may be the main reason why their abundance significantly negatively correlated with MSUBSTD (Figure 5B). In addition, annual precipitation may be the most important environmental factor for the difference in community composition between the two basins (Figure 5B). Annual precipitation may affect macroinvertebrates communities by changing the hydrological conditions [66,70,71]. Some species, such as Calopterygidae and Ephemerellidae, that require fast-flowing waters showed significant positive correlations with annual precipitation (Figure 5B) and were more abundant in the Dongjiang basin where the headwater streams covered high elevation range. The headwater streams in the Dongjinag basin are located on the east side, and receive higher annual precipitation, which is a driving force to generate substrate with larger particle size (i.e., low MSUBSTD).

## 5. Conclusions

Our study presents a case of macroinvertebrates diversity and community assembly in headwater streams in subtropical lowland basins, which share a species pool. The two groups of studied headwater streams are located on two sides of a mountain. In contrast to our hypothesis, shading in East Asia monsoonal climate, different environmental conditions, and taxon composition have been built between the neighboring basins located on the two sides of a mountain. Highly similar taxa between the two neighboring basins demonstrated that environmental filtering is similar between the two neighboring basins at the regional scale. However, basin identity needs to be considered at a large scale to better predict the responses of macroinvertebrates to stressors. Environmental selection plays an important role in community assembly within individual basins, in which the substrate heterogeneity and chemical variables of headwater streams are the most significant explaining variables.

**Supplementary Materials:** The following supporting information can be downloaded at: https://www.mdpi.com/article/10.3390/d14050402/s1, Figure S1. PCA plots of chemical variables (A), physical variables (B), and climatic variables (C); Table S1. Correlations between the first two PCA axes (PC1 and PC2) and chemical variables; Table S2. Macroinvertebrate composition and abundances in streams of the Beijiang basin (BJ1–BJ9) and Dongjiang basin (DJ1–DJ8).

**Author Contributions:** Conceptualization, L.W. and B.-P.H.; methodology, data collection: L.W., J.L., L.T. and X.L.; writing—original draft preparation, L.W., E.Z.R. and B.-P.H.; writing—review and editing, E.Z.R. and B.-P.H.; funding acquisition, B.-P.H. All authors have read and agreed to the published version of the manuscript.

**Funding:** This research was funded by the grant for NSF of China, grant number 32171538.

**Institutional Review Board Statement:** Not applicable.

**Informed Consent Statement:** Not applicable.

**Data Availability Statement:** Not applicable.

**Acknowledgments:** The authors acknowledge the National Natural Science Foundation of China, grant number 32171538. The authors are thankful for the help of colleagues and students from the Department of Ecology and Institute of Hydrobiology in Jinan University during the field work. Henri Dumont from Ghent University in Belgium is appreciated for reading.

**Conflicts of Interest:** The authors declare no conflict of interest.

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
