# Peer review of "Species Diversity and Community Composition of Macroinvertebrates in Headwater Streams of Two Subtropical Neighboring Lowland Basins"

_diversity, doi:10.3390/d14050402_

Round 1

Reviewer 1 Report

Dear Authors,

I read with interested your manuscript but, I am sorry to say that I don't agree with your conclusion and how you intrepretad your results.

In general I think you trusted your statistical results too much and you sometimes missed the ecological significance between them. In my opinion,  when you obtain some significant results (p<0.05) it doesn't mean that you have to drive your conclusion only on the base of them. I am refering for example on the MDS stress or the ANOSIM (lines 204-25) or the results reported from the lines 230.

I am sorry but I think that your conclusion are driven more by statistical results than ecology. 

Some suggestions:

lines 204-5 and figure 4: statistical significance (stress) of the MDS and ANOSIM test result are not so high so your conclusion are supported by statistic but ecologically too weak

lines 217-220: please add the p-values

line 230: Apparently the environmental factors don't explain a very high variance. Is it enough for your conclusions?

line 259: you mentioned only 9 streams of Dongjiang. What about the other river?

line 266: you campared species diversity with family diversity in teh same context. Please better define this concept

lines 273-274: maybe it should be interesting to add the number of specimens you collected in order to reinforce your results

line 302: a difference statistically  significant but less ecological as you stated in the following sentence

Reviewer 2 Report

Review

Paper title: Species diversity and community composition of macroinvertebrates in headwater streams of two subtropical neighboring lowland basins

The authors conducted a comparative study to reveal the significance of environmental factors in driving the species diversity and community composition patterns of macroinvertebrates from two rivers in China. They found that the basin identity was the key factor explaining the variation of communities. Annual precipitation was also demonstrated to be a driver in distinctive substrate compositions in the two basins. These results provide a baseline for further studies and expand the knowledge of macroinvertebrate communities in subtropical freshwater systems.

All these reasons explain the relevance of the paper by Lu Wang and co-authors submitted to "Diversity".

General scores.

The data presented by the authors are original and significant. The study is correctly designed. In general, the statistical analyses are performed with good technical standards. We authors conducted careful work which will attract the attention of a wide range of specialists focused on macroinvertebtare biology in the freshwater environment.

Recommendations.

Figure 2. The authors should indicate the kind of error (vertical bars) used here (SE, SD, 95%C.I.). The authors should include the units for the measured parameters on the OY-axes.

Figure 4. The authors should increase the font size and resolution.

Figures 6, 7. The authors should provide the full description of codes used in this figure.

The authors should update the discussion by considering the following papers:

Paller, M.H.; Blas, S.A.; Kelley, R.W. Macroinvertebrate taxonomic richness in minimally disturbed streams on the Southeastern USA Coastal Plain. Diversity 2020, 12, 459. https://doi.org/10.3390/d12120459

Farooq, M.; Li, X.; Li, Z.; Yang, R.; Tian, Z.; Tan, L.; Fornacca, D.; Li, Y.; Cili, N.; Ciren, Z.; Liu, S.; Xiao, W. The joint contributions of environmental filtering and spatial processes to macroinvertebrate metacommunity dynamics in the alpine stream environment of Baima Snow Mountain, Southwest China. Diversity 2022, 14, 28. https://doi.org/10.3390/d14010028

Brysiewicz, A.; Czerniejewski, P.; DÄ…browski, J.; Formicki, K. Characterisation of benthic macroinvertebrate communities in small watercourses of the european central plains ecoregion and the effect of different environmental factors. Animals 2022, 12, 606. https://doi.org/10.3390/ani12050606

Supplementary material.

Table S2. The authors should add units. Figure S1. The authors should provide the full description of codes used in this figure.

Specific comments.

L 46. Change “macroinvertebrates” to “macroinvertebrate communities”

L 57. Change “high hydrological fluctuation” to “substantial hydrological fluctuations”

L 58. Change “influence in” to “influence on”

L 68. Change “covers” to “cover”

L 161. Change “was conducted” to “were conducted”

L 171, 182, 184, 189, 209, 217, 218, 246, 249, 254, 290. Change “Beijiang basin” to “the Beijiang basin”

L 183, 185, 208, 218, 247, 251, 259, 290. Change “Dongjiang basin” to “the Dongjiang basin”

L 188. Change “significant difference” to “significant differences”

L 203. Change “the Chironominae” to “Chironominae”

L 233. Change “were negligible” to “was negligible”

L 237. Change “factors affecting” to “drivers of”

L 238. Change “explained” to “explaining”

L 244. Change “basin model” to “the basin model”

L 255. Change “Dongjiang basin” to “and the Dongjiang basin”

L 257. Change “the separate sides” to “separate sides”

L 273. Change “collected” to “recorded”

L 293. Change “linked with” to “linked to”

L 299. Change “high” to “there is high”

L 308. Change “little difference” to “a little difference”

L 316. Change “but by the” to “but the”

L 329. Change “Environmental” to “The environmental”

L 329. Change “explained” to “the explained”

L 333. Change “macroinvertebrates” to “macroinvertebrate”

L 335. Change “significantly positively correlation” to “significant positive correlations”

L 336. Change “be  more” to “were  more”

L 337, 344. Change “Dongjiang basin” to “the Dongjiang basin”

L 338. Change “At basin scale” to “At a basin scale”

L 339. Change “Beijiang  basin” to “Beijiang  basin”

L 352. Change “regional  scale” to “a regional  scale”

L 354. Change “significant difference” to “the significant difference”

L 356. Change “at large scale” to “at a large scale”

Reviewer 3 Report

This manuscript presents the results of a short term field study (2 to 3 weeks of sampling) involving sampling macroinvertebrates and habitat variables from 17 sites in two subtropical headwater streams in China. Below I provide comments on the manuscript that I hope will help the authors with improving the manuscript. 

Overall Comments:

A. line 69-71: Why do you hypothesize that species composition and community structure will be similar if environmental filtering is the major influencing factor? The theoretical and/or quantitative support for this hypothesis has not been provided within the Introduction.  Additionally, the concept of environmental filtering (Tonn et al.1990. American Naturalist; Poff 1997. Jour N. Amer. Benth Soc) indicates that that large scale factors serve to determine the species composition and local scale factors determine the fine scale diversity patterns.  As such I would have expected a hypothesis similar to the following: we hypothesize that if environmental filtering is important then we would observe that macroinvertebrate species composition would be more influenced by large scale regional factors (i.e., basin, climate) than local environmental factors (i.e., nutrients, water velocity, substrate) and that macroinvertebrate diversity would be more influenced by local environmental factors than large scale regional factors.  However, that suggestion is simply my perspective from what I understand about the theory surrounding environmental filtering in community ecology.  I suggest the authors revisit the citations I suggested above as well as others (papers by Keddy and coauthors)  to further refine their hypotheses.   

B. In reviewing the supplementary file and Table S2 I note the use of inappropriate use of morphospecies (i.e., Diptera sp. 1, Ditera sp. 2, Limnephilidae sp 1, Limnephilidae sp 2) and because the lowest taxonomic level of identification here is the Order level for Diptera and Family level for Limnephilidae, and as such Dipt sp1 and Dip sp 2, should be combined into simply Diptera and Limnephilidae sp 1 and Limnephilidae sp. 2 should be combined into one taxa Limnephilidae. This needs to be done for all subsequent taxa that suffer from this to ensure that diversity is appropriately calculated in this manuscript because as such the manuscript over inflates macroinvertebrate diversity.   Addressing this comment will require redoing all diversity calculations and all subsequent statistical analyses. 

C. line 146 and throughout the rest of the manuscript – because you have a mixed level of taxonomic resolution it is inaccurate to indicate that you calculated species richness, especially since none of the taxa were identified to species level. Instead it is more accurate to state you calculated “taxa richness” and need to use this term instead of “species richness” in all instances in the manuscript. Same logic follows for “species composition” … instead use “taxa composition” throughout the entire manuscript.

D. Statistical analyses: The authors conducted a large number of statistical analysis involving the Mann-Whitney Test, t-tests, PCA, RDA, NMDS, ANOSIM, additive partitioning of rich and abundance, Jaccard and Ruzicka dissimilarity indexes and subsequent beta replacement and differences. The problem is that the authors did not clarify how each test helps them address their hypothesis of interest and without the link being made between each statistical test and the hypothesis the readers do not know how the test results contribute to understanding if the authors findings supported or did not support their hypotheses.  If the authors find that they cannot clearly link a particular test to their hypotheses, then that test should not be part of the statistical analysis for this paper.  Additionally, some analyses (PCA, RDA) were conducted for both headwater streams combined and for each headwater stream alone.  I question the value of conducting these analyses for each headwater stream alone because of the sample sizes from each stream (9 in one and 8 in another) are so small that the observed environmental relationships are not likely robust.  Again, it is not clear to me how doing the analyses for each stream helps address the hypothesis.  Normally, as part of my review if I disagreed with the statistical analysis approach I would provide a recommendation on how to revise the analyses.  However, I do not provide a recommendation because I cannot because the authors hypothesis is not clear and it is potentially inaccurate.    

Specific Comments:

line 14-15: revise as follows:  ….sampled macroinvertebrates and measured habitat variables in…

line 19: write out NMDS and avoid use of acronyms in the abstract

line 26-27:  replace “macroinvertebrate” and “subtropics” as keywords because these words are in the title

line 29-30:  exactly what related to community composition and mechanisms of species diversity is a central tenet of community ecology?  be specific

line 46:  what does rich macroinvertebrates mean?   do you mean that the taxonomic richness of macroinvertebrates is high?  if so state that

line 55-57:  While I could be wrong I thought it has been established that macroinvertebrate diversity is greater in the tropics than temperate regions, at least for terrestrial invertebrates. if this is not true for aquatic invertebrates then need to provide supporting citations for this statement and clarify the distinction in the patterns between terrestrial macroinvertebrates and aquatic macroinvertebrates

89-91:  why only sample 9 sites in one basin and 8 sites in another basin?  The low sample size is surprising given that each site was only sampled one time.  Need to provide an explanation why these 17 sites were selected and other sites were not sampled.  

line 102:  why were these 17 sites only sampled during one season in one year?  I understand the value of sampling during the winter when discharge and stream depths are low enough to allow sampling, but why were these sites only sampled in one year, why not for two or three years, which would have increased the sample size and improved the robustness of the results.  Need to clarify the reason for sampling these 17 sites only once.  

line 105 to 106:  need to clearly communicate how the surber samples were allocated among microhabitat within a site.   Simply stating usually 3 riffles and 2 pools is not acceptable.  How many sites did you sample 3 riffles and pools?  what other variations occurred – did you sample 4 riffles and 1 pool? if so how many sites.  Did you sample 3 pools, 1 run, and 1 glide?  if so how many sites were sampled in that manner.  Be specific here.   If the proportion of sampled microhabitats differed greatly among sites then that could be an underlying factor that would influence the observed habitat relationships. 

line 108:  simply stating “lowest possible taxonomic level (usually genus) is not appropriate.  Another scientist could not repeat your study with this information.  Additionally, the supplementary file indicates a mixed taxonomic level was used with different taxonomic levels for different taxa.  This is the norm among macroinvertebrate studies.   Describe clearly, which taxa identified to genus, which identified only to family, and which identified only to order. 

line 114: how many measurements of reach width were made?  usually stream ecologists measure reach width or the width of the wetted portion of the stream every 20 m or so (6 times in a 100 m reach.

line 114: how many measurements of velocity were made?  usually stream ecologists measure between 2 to 10 velocity measurements among multiple transects within a site. 

line 114:  why was water depth not measured?  This is an important variable and one that is easily measured.  I am surprised it was not measured and the authors need to explain here why it was not measured.

114-125:  when were environmental variables measured?  where they measured on the same day that the sites were sampled for macroinvertebrates?  if so specify that .  If not clarify when the environmental variables were measured. 

line 129:  which environmental variables were compared between the two streams?  be specific

line 130: why was non-parametric Mann-Whitney used for environmental variable comparisons?  I assume it was because the data were either non-normal or did not meet assumptions of equal variance, but the authors need to clarify this for the readers along with an explanation of the tests they used to determine that the data were not normal and/or exhibit equal variance. 

138-139:  this information belongs in the Results section not the Methods section

line 141:  what is “8SILTCLAY”?  should this be “8 X SILTCLAY

line 141 -145: need to define TOTSUB and describe how it was calculated.  This is integral to understanding this formula. 

line 147:  was normality and equal variance of taxa richness and density confirmed?  if so, which tests were used to do so?  Need to report that here.  Also, what program was used to conduct t-tests? 

line 168: what variables represent spatial process.  Need to define this term and explain what variables were used to represent it. 

line 175 and all other instances:  delete the “” associated with the names of the R packages. 

line 183:  need provide test statistic and p values for DO and pH results

line 184: need provide test statistic and p values for MSUBSTD results

line 185: need provide test statistic and p value for annual precipitation results

Figure 2, lower right subfigure – need to make the y axis intervals consistent (0, 200, 400, 600, 800, 1000, etc.., instead of the unequally spaced intervals used (0, 400, 1600, 2000)

line 191:  which taxa were similar between the two streams?

line 197: report actual p value here

line 202 and throughout manuscript:  if only identifying animals to genus level then do not need “sp.1” following the genus name.  Same follows for those animals identified to order or family level. 

line 205:  replace “showed’ with “indicated”

line 244:  the residual values are presented in the center of the figure not the lower right corner

line 275-277:  these are meaningless comparisons because the taxonomic composition from which the richness values were derived is not shared among all studies.  For example in this study used a mix of genus, family and order level and the taxa richness observed here is not comparable to another study that calculated taxa richness based on Family level identifications.  need to revise to confirm for the reader that the comparisons being made are those that are valid comparisons. 

line 283:  insert “because of” between “sources” and “high”

line 284:  delete “could help explain this phenomenon”

line 285: replace “are” with “were”

line 290: how is 12.16% and 10.61% considered a high proportion of the predators? 

line 316: according to the information in the results the spatial model did not explain any aspect of the variation of macroinvertebrate communities

Round 2

Reviewer 1 Report

Dear Authors,

despite you modified the manuscript and answered my comments, I still remain with my idea.

I am sorry but I think that this article is not suitable for the publication

Best regards

Author Response

1) We do our best to improve the English language and style;

2) We also improved our statements of statistical conclusions so that to match the ecology.

Reviewer 3 Report

This manuscript is a revised version of one I reviewed previously that presents the results of a short term field study (2 to 3 weeks of sampling) involving sampling macroinvertebrates and habitat variables from 17 sites in two subtropical headwater streams in China. Below I provide comments on the manuscript that I hope will help the authors with improving the manuscript. 

Overall Comments:

A. First, I still question the validity of the authors’ use of morphogenus designations. I understand that in many parts of the world the taxonomy of aquatic insects has not been delineated.  However, this is an accuracy issue – if the animals taxonomy has not been refined to enable taxonomic identification to genus, then how can one confidently know that the “morphogenus designation” reflects a genus? One cannot.  In my view it would be more accurate simply to consider these taxa at the level they could be identified be it subclass, suborder, family, etc.  The use of a mixed taxonomic resolution and taxa richness enables taxa richness to be counted based on any level of identification. Let’s consider the following examples with information derived from Table S2:

- In those 24 cases were only one higher level taxa exists (Hydropsychidae g1, Polycentropodidae g1, Glossomatiidae g1, Ecnomidae g1, Leptoceridae g1, Tabanidae g1, Ceratopogonidae g1, Dixidae g1, Heptagniidae g1, Isonychiidae, Siphlonuridae, Leptophlebiidae, Potamanthidae, Chloroperlidae, Taeniopterygidae, Perlodidae, Calopterryfidae, Corydalidae, Gomphidae g1, Cordulegastridae, Libellulidae, Psephenidae, Dytiscidae, Gammaridea) – assigning these as genus level does not result in overcounting taxa richness because only one taxa is identified. 

-additional Table S2 lists the 7 taxa in which multiple morphogenus taxa were identified (Limnephilidae g1, 2; Psychomyiidae g1, 2; Tipulidae g1, 2, 3, 4, 5, 6, 7, 8; Diptera g1, g2; Coleoptera g1, 2, 3, 4; Elmidae g1, g2; Oligochaeta g1, g2).  For these seven groups taxa richness potentially is overcounted.  Using the current approach taxa richness from these 7 taxa and all their subsequent morphogenuses would be 20 as compared to a taxa richness 7 if it was only calculated based on the actually identification level made, which corresponds to a 185% increase in the estimate of taxa richness from just these 7 groups. 

Additionally, I still maintain that morphogenus should not be used because it results in inaccurate descriptions of the taxonomic resolution and potentially overestimates diversity.  Ideally, the authors would exclude morphogenus identifications and use the identifications they have in hand and are all valid identifications.   The only taxa where recalculations need to be remade would be the 7 taxa listed above that need to be collapsed into one taxa for abundance and richness calculations and comparisons of taxa composition.  I understand the authors’ hesitancy to recalculate the response variables and redo all of the statistical analysis.  Doing so would create a stronger and more accurate paper.  However, at the very least the authors need to accurately state their taxonomic resolution used, to clearly identify the groups with morphogenus designations and potentially uncertain taxonomic identifications and present the information in Table S2 accurately.  If the authors decide only to accurately state their taxonomic resolution and clearly identify the groups with morphogenus designations then the following changes need to be made:

line 124-125:  this sentence is inaccurate because all those morphogenus designations are not genus level identifications.   Need to revise this sentence to indicate which taxa were actually identified to genus level, which taxa were assigned morphogenus identifications based on the authors’ taxonomic judgement, and the Chironomidae taxa identified to subfamily. 

line 124-126:   Need insert sentence or two here specifically identifying those taxa in which the genus was unknown but a morphogenus was assigned along with explanation for doing so (i.e., many genus are undescribed and that the morphogenus was described based on the authors’ taxonomic judgement)

B. Need to provide explanation for low sample size: In their response letter the authors indicate that one reason that they only sampled 8 or 9 streams in each basin was because in their study they wanted to focus on undisturbed headwater streams. This is an important detail that needs to be shared in the paper because it provides information about the sampled streams and the context to which the observed results apply (undisturbed subtropical streams in China).  Thus, I recommend revising line 103 as follows:

 ..in the Dongjiang basin.  Our focus for this study was to sample undisturbed headwater streams and as such the sampled streams were the only ones we could find. The two groups..

C. Statistical analyses: The description of the statistical analysis has improved and I commend the authors’ for their work on this. However, it is not clear to me as a reader why the PCA analyses and RDA analyses were conducted for both headwater streams combined and for each headwater stream alone.  I question the value of conducting these analyses for each headwater stream alone because of the sample sizes from each stream (9 in one and 8 in another) are so small that the observed environmental relationships are not likely robust.  Additionally, although the authors need to confirm it I suspect that the two basin PCA model provided the independent variables used to represent the local scale environmental model in the two basin RDA.  It is also clear to me that the two basin model RDA contributes to the hypothesis because it identifies the relative influence of the environmental model, the basin model, and the spatial model.   However, the single basin RDA results are not used to support the hypothesis and they are only discussed to share some observations related to the habitat relationships within the individual basins in the Discussion sections.  I strongly recommend deleting the individual basin PCA and individual basin RDA from the analyses.  Doing so will simplify the description of the statistical analysis approach and presentation of the results so it focuses on the most important results related to the hypotheses. 

Specific Comments:  

line 121:  replace “subsites:”with “microhabitats”

line 137:  need specify where the 3 transects were located in a site and how far apart they were from each other

line 166: as explanatory variables in which analyses?  need to be specific.  was it the RDA analyses?  As written this is not clear

line 170:  replace “TOTSBU” with “TOTSUB”

line 199:  replace “total” with “measured”

line 201-202:  was the create.MEM.model function used to create the spatial model mentioned in the previous sentence?  if so need to reword this sentence to clearly indicate that for the reader

line 201:  Also is the create.MEM.model function an R function?  if so what package is it from?  I searched for this function in google and could not find it.  I then went to citation 23 that is listed in association with this and that citation indicates that they used the MEM spatial eigenfunctions were computed using the PCNM function of the PCNM package for their calculations. Admittedly, I might have missed something in my searches for this information, but additional information needs to be added here to resolve the identified inconsistencies.   

line 209:  the finding that the spatial model was not significant belongs in the results section not the methods section.

line 213:  need to specify what the full RDA model is.  Is this the RDA run with data from both basins together or is it the RDA run with all independent variables? 

line 239 and Figure 3:  Replace Figure 3 with Table S3,  The results presented in Table S3 belong in the paper and are more important than Figure 3.  Once this is completed then revise line 239 as follows:  …two basins (Table 1).   Note Tables that follow will need to be renumbered.

line 244 and throughout the manuscript:  delete “sp.” following genus  and subfamily names

line 311:  revise as follows:  …of a mountain and we observed differences in selected environmental variables between basins likely influenced by east Asian monsoons.  The…

line 364:  replace “gene level” with “at the level of our taxonomic resolution (mostly genus level), the….

line 376:  which RDA analysis the two basin model or the single basin models? 

line 403: there is no Figure S2 in the supplementary file

Table S1.  Need to indicate in the table legend what the “*” and “**” next to the correlation coefficients mean. 

Table S2 supplementary file:  Remove “sp.” following those taxa that have been identified to genus and the chironomidae identified to subfamily.  The genus and/or subfamily names alone in these instances will suffice.  The use of “sp” after these names inappropriately implies a species level designation.  

Table S2 supplentary file:  Table 2 still contains 1 suborder (Gammaridea), 1 Subclass (Oligochaeta) and 13 families (Isonychiidae, Siphlonuridae, Leptophlebiidae, Potamnthidae, Chloroperlidae, Capniidae, Taeniopterygidae, Calopteryfidae, Corydalidae, Cordulegastridae, Pentatomidae, Psephenidae, Dytiscidae) with inappropriate morphospecies designations. 

Table S2 supplentary file:  For those taxa that represent morphogenus need insert explanation in the table legend explaining the designation. 

Author Response

Appreciate your valuable comments again. Please see our responses in the attachment. All line numbers refer to the line numbers in the track change version.

Round 3

Reviewer 3 Report

Good job on making the revisions to this manuscript.  It is much improved. 

Author Response

Thanks for your suggestion. We have carefully proofread and corrected our manuscript.